# Marker-Assisted Selection of Jacalin-Related Lectin Genes *OsJRL45* and *OsJRL40* Derived from Sea Rice 86 Enhances Salt Tolerance in Rice

**DOI:** 10.3390/ijms252010912

**Published:** 2024-10-10

**Authors:** Xiaolin Yin, Qinmei Gao, Feng Wang, Weihao Liu, Yiting Luo, Shuixiu Zhong, Jiahui Feng, Rui Bai, Liangbi Chen, Xiaojun Dai, Manzhong Liang

**Affiliations:** Hunan Province Key Laboratory of Crop Sterile Germplasm Resource Innovation and Application, College of Life Science, Hunan Normal University, Changsha 410081, China; yinxleffort@163.com (X.Y.); 15802620355@163.com (Q.G.); 13975877142@163.com (F.W.); 18749958161@163.com (W.L.); luoyiting2023@163.com (Y.L.); zsx13207975104@163.com (S.Z.); jiahuifeng@hunnu.edu.cn (J.F.); bairui0511@163.com (R.B.); chenliangbi@126.com (L.C.)

**Keywords:** rice (*Oryza sativa* L.), *OsJRL45*/*OsJRL40*, recombinant inbred lines (RILs), transcriptome

## Abstract

Soil salinization limits rice growth and is an important restriction on grain yield. Jacalin-related lectins are involved in multiple stress responses, but their role in salt stress responses and use as molecular markers for salt tolerance remain poorly understood. Salt stress treatments and RT-qPCR analyses of Sea Rice 86 (SR86), 9311, and Nipponbare (Nip) showed that *OsJRL45* and *OsJRL40* enhanced tolerance of salt stress in SR86. Molecular markers based on sequence differences in SR86 and the salt-sensitive variety, 9311, in the intergenic region between *OsJRL45* and *OsJRL40* were validated in recombinant inbred lines derived from SR86 and 9311, hybrid populations, and common rice varieties. Yeast two-hybrid and bimolecular fluorescence complementation demonstrated that OsJRL45 and OsJRL40 interacted. Co-transformation of Nip with *OsJRL45* and *OsJRL40* derived from SR86 had no effect on the mature phenotype in T_2_ plants; however, salt stress at the three-leaf stage led to significant increases in CAT, POD, SOD, and Pro contents, but reduced MDA content in transgenic plants. Transcriptomic analysis identified 834 differentially expressed genes in transgenic plants under salt stress. GO and KEGG enrichment analyses indicated that metabolic pathways related to antioxidant responses and osmotic balance were crucial for salt-stress tolerance. Thus, molecular markers based on nucleotide differences in *OsJRL45* and *OsJRL40* provide a novel method for identifying salt-tolerant rice varieties.

## 1. Introduction

Rice (*Oryza sativa* L.) is one of the most important food crops globally, with over half of the world’s population relying on it as a staple food [1]. The demand for food consumption is rising due to the rapid increase in the global population, although the area of land suitable for arable production decreases year by year due to the increase in salinized land [2]. Addressing the gap between this increased demand due to population growth and the reduction in arable land presents a significant challenge for grain production in China [3,4]. Rice is a salt-sensitive crop that, when subjected to salt stress, responds at the morphological, cellular, and molecular levels [5] in ways that affect essential processes, including photosynthesis and metabolism, leading to growth inhibition or even death [6,7]. At the molecular level, salt stress leads to ion toxicity, osmotic stress, and oxidative stress [8]. Ion toxicity under salt stress is usually due to the excessive accumulation of Na^+^ and Cl^−^, which causes Na^+^ to accumulate in the cytoplasm. This leads to membrane depolarization and K^+^ leakage from the cell, ultimately resulting in plant death [9]. OsHAK1, a potassium transporter, promotes K^+^ uptake under salt stress and increases the K^+^/Na^+^ ratio, making this protein crucial for enhanced salt tolerance in rice [10,11,12]. Many other genes related to ion transport and the regulation of salt tolerance have been identified, including *OsHKT1;5* [13] and *OsAKT2* [14]. To cope with the osmotic stress caused by salt stress, plants produce large amounts of osmotic regulators, including proline, glycerol, sugars, and their derivatives [15]. Salt stress-induced oxidative stress increases the activity of antioxidant enzymes such as catalase (CAT), peroxidase (POD), and superoxide dismutase (SOD), as well as the synthesis of some non-enzymatic antioxidants [16]. The salt tolerance mechanism is thus a complex regulatory network composed of multiple metabolic pathways.

Increased soil salinity necessitates the development of novel methods for improving the tolerance of existing rice varieties to salt stress, as well as the use of conventional breeding techniques, such as backcrossing, recurrent selection, and use of mutations, to develop new varieties. Such approaches have their limitations, as, in addition to the desired genes, unwanted genes may also be passed on to future generations [17]. As genetic diversity serves as the raw material for breeding programs, the generation of segregating populations (F_2_ progeny or recombinant inbred lines (RILs)) followed by genetic analysis using a conventional genetic mapping approach can identify important agronomic traits and genes under controlled conditions [18,19,20]. Such populations contain a large number of genetic variants, enabling advances in crop improvement and the genetic analysis of complex traits, as well as the development of genetic markers to detect quantitative trait loci and isolate the genes responsible [21]. Marker-assisted selection (MAS) is an important molecular tool for rice breeding that minimizes the number of backcross generations required [22]. Markers are used to select indirectly the genetic determinant or determinants of a trait of interest, such as abiotic stress tolerance [23]. The development and availability of markers for use with plant lines carrying genomic regions known to participate in the expression of target traits has made it possible to use MAS to identify major genes controlling such traits [24].

Lectins are proteins that are found in a wide variety of species, including microorganisms, plants, and animals [25]. In plants, lectins are found in vegetative organs such as roots, stems, and leaves [26]. *HvHorcH* is a member of the jacalin-related lectin (JRL) family that is involved in the physiological response of barley (*Hordeum vulgare*) roots to salt stress and also enhances salt tolerance [27]. Abscisic acid (ABA) biosynthesis is inhibited in transgenic Arabidopsis (*Arabidopsis thaliana*) plants expressing *PeJRL* (*Populus euphratica*), thereby reducing the production of ABA-induced ROS and oxidative damage during salt stress [28]. OsJRL, a rice protein, is a JRL belonging to the mannose-binding lectin superfamily. It is up-regulated by *OsJRL* overexpression and increases the expression of stress-related genes under salt stress, enhancing salt tolerance [29]. Lectins are indirectly involved in regulating chlorophyll content and also reduce membrane damage and electrolyte leakage, thus mitigating the effects of salt stress [30]. The expression of *SIT1*, a lectin-encoding gene in rice, is up-regulated under salt stress; SIT1 participates in the MPK3/MPK6 pathway, which increases ethylene synthesis and ROS accumulation [31], and also plays a role in Na^+^ efflux by regulating the salt overly sensitive (SOS) pathway [32]. Under salt stress, rice plants overexpressing *OsSalT*, which encodes a mannose-binding protein, have significantly lower Na^+^ content than wild-type plants, indicating that this gene plays an important part in the response to salt stress [33]. *OsJRL45* enhances the antioxidant capacity of rice plants and promotes the maintenance of Na^+^-K^+^ homeostasis under salt stress; *OsJRL45*-overexpression lines show higher salt tolerance, as well as significantly increased grain-filling rates and thousand-grain weights [34]. *OsJRL40* enhances antioxidant enzyme activity and maintains Na^+^-K^+^ homeostasis under salt stress, as well as regulating salt tolerance in rice by controlling the expression of Na^+^-K^+^ transporters, salt-response transcription factors, and other salt response-related proteins [35]. Lectins thus play key roles in regulating stress response pathways during salt stress; however, their specific mode of action in response to salt stress requires further study.

The Sea Rice 86 (SR86) cultivar was developed over years of selection and offers many advantages, including high salt tolerance [36]. The identification of OsNACL35, a NAC (NAM, ATAF1/2, and CUC2) transcription factor, in SR86 suggests that salt tolerance may result from an enhanced H_2_S yield [37]. Microbial metabolomic analysis of rhizosphere bacteria associated with SR86 seedlings cultivated under different levels of salinity revealed that salt stress significantly alters the diversity of rhizosphere bacteria and rhizosphere metabolites, thereby improving salt tolerance [38]. The *OsJRL45* and *OsJRL40* genes from SR86 enhance the antioxidant capacity of rice plants and promote the maintenance of Na^+^-K^+^ homeostasis under salt stress [34,35,39]. Not all of the genes related to salt tolerance in SR86 have been identified to date, and most of those that have need to be expressed under stress induction. The use of molecular markers, linkage analysis, and mapping techniques will enable the identification of more genes involved in salt tolerance in this cultivar.

Our previous study hybridized the salt-tolerant rice variety SR86 with a standard cultivar Nip and used bulked segregant analysis (BSA) and genetic mapping to identify salt-tolerant and salt-sensitive populations in the segregating hybrids. We then screened these populations for genes related to salt stress and identified two JRL genes, *OsJRL45* and *OsJRL40* [39]. We also verified the physiological and biochemical functions of *OsJRL45* and *OsJRL40* in salt tolerance in rice [34,35]. In the current study, to better understand the roles played by *OsJRL45* and *OsJRL40* in salt tolerance, and to develop a novel method for mapping salt tolerance in rice, we co-transformed the Nip cultivar with *OsJRL45* and *OsJRL40* from SR86, and also constructed a segregating population derived from SR86 and 9311, a salt-sensitive variety. We screened the hybrid population for salt-tolerant plants by examining the phenotype under salt stress. We identified nucleotide differences associated with salt tolerance in the segregating population and used these to develop molecular markers for salt tolerance. This method improved the efficiency of screening for salt tolerance and shortened the breeding process of salt-tolerant rice, promoting the rapid development of new salt-tolerant rice varieties.

## 2. Results

### 2.1. Analysis of OsJRL45 and OsJRL40 Expression under Salt Stress

To analyze the expression of *OsJRL45* and *OsJRL40* in SR86, 9311, and Nip under salt stress, seedlings at the three-leaf stage were treated with 9‰ NaCl for 4 days, followed by a 4-day recovery period (Figure 1A–C). The survival rates of 9311 and Nip were around 20%, while the survival rate of SR86 was 80%, indicating that this variety had significantly higher salt tolerance than either 9311 or Nip (Figure 1D). RT-qPCR analysis of *OsJRL45* and *OsJRL40* expression in SR86, 9311, and Nip after 6, 8, 24, and 48 h of salt stress revealed that both genes were expressed at significantly higher levels in SR86 than in 9311 and Nip (Figure 1E,F). These results suggested that *OsJRL45* and *OsJRL40* enhanced salt stress tolerance in SR86.

### 2.2. Analysis of Sequence Variation in OsJRL45 and OsJRL40

*OsJRL45* and *OsJRL40* are located on rice chromosome 4 and are thus likely to interact with each other (Figure 2A,B). An analysis of sequence variation between SR86 and other rice cultivars revealed a correlation with salt tolerance. The *OsJRL45* and *OsJRL40* sequences from Shuhui 498 (R498) were obtained from the NCBI (https://www.ncbi.nlm.nih.gov/, accessed on 29 June 2022). Both genes contained two exons and one intron (Figure 2). Primers were designed to amplify the *OsJRL45* and *OsJRL40* sequences from the rice cultivars SR86, 9311, L623, R498, Minghui63, Zhenshan97, Yu Chi, Nantehao, Qingzhen, and Meizhan, using NCBI R498 as the reference sequence.

Sequence alignment showed that exon 1 of *OsJRL45* contained eight single nucleotide polymorphisms (SNPs). Amino acid alignment revealed that three SNPs (located 63 base-pairs (bp), 72 bp, and 123 bp after the start codon) were synonymous mutations but that the SNPs located 209 bp, 224 bp, 245 bp, 263 bp, and 268 bp from the start codon led to amino acid changes (Figure 2B). Exon 2 of *OsJRL45* contained one SNP in which adenine (A) in the SR86 and Nantehao sequences was substituted with cytosine (C) in the other cultivars, leading to a change from Glu to Asp in the amino acid sequence (Figure 2B). An SNP in exon 1 of *OsJRL40* was found in a single cultivar, Meizhan. Exon 2 of *OsJRL40* contained one SNP; an adenine (A) in the SR86 sequence was substituted with guanine (G) in other cultivars, resulting in an amino acid change from Ser in SR86 to Gly in the other cultivars (Figure 2B). Genomic sequencing of *OsJRL45* and *OsJRL40* from SR86, R498, Minghui63, and Zhenshan97 revealed that the two genes were only 2605 bp apart. In SR86, the intergenic region contained a 25-bp deletion (GCGCACGACGCGGCCGCCAGCGTTC); this deletion was located 204 bp downstream from the end of the *OsJRL45* coding sequence and 2401 bp upstream of *OsJRL40*. We therefore designed and validated molecular markers based on the variation in the *OsJRL45* and *OsJRL40* sequences in the different rice varieties.

### 2.3. Validation of Molecular Markers in Populations

An F_1_ generation of hybrid progeny was produced by crossing SR86 and 9311. Individual F_1_ plants were selected and self-pollinated to produce an F_2_ population. We selected 3000 F_2_ seeds, which were germinated and the resulting seedlings subjected to 9‰ NaCl salt stress at the three-leaf stage. The surviving salt-tolerant plants were allowed to grow until seed-set and harvest. We then selected 320 individual F_3_ seedlings, which were grown to the three-leaf stage under hydroponic conditions, treated with 6‰ NaCl for 5 days to induce salt stress, and then allowed to recover for 5 days (Figure 3). DNA was extracted and analyzed by PCR using the molecular marker primers developed in this study. The PCR products were separated using 2% agarose gel electrophoresis (Appendix A), and a statistical analysis was performed to compare individual plants in the population. Of the F_3_ plants tested, 228 plants produced an agarose gel electrophoresis result that closely resembled that of the SR86 parent, 54 plants produced a result that resembled that of the 9311 parent, 36 plants were heterozygous, and the remaining 2 plants did not produce any PCR product (Table 1). These results indicated that molecular marker primers based on nucleotide differences between *OsJRL45* and *OsJRL40* could be used to screen salt-tolerant plants.

### 2.4. Validation of Molecular Markers for Salt Tolerance

To validate the accuracy of the molecular markers developed in this study to screen for salt-tolerant rice, the markers were verified in a number of rice varieties. DNA was extracted, amplified by PCR, and the PCR products separated using 2% agarose gel electrophoresis. The rice varieties that showed higher salt tolerance (Foer B, Chaimi, Yuejing, Xiangzao 32, Nantehao, R288, Shuhui 527, Aizinuo, and Xiang 628S) produced gel electrophoresis results that closely resembled that of SR86, while salt-intolerant plants produced similar results to 9311 (Figure 4). This indicated that the novel molecular markers designed to identify salt tolerance were accurate for more than 90% of plants, suggesting that they had high specificity when used for screening.

### 2.5. Validation of MAS for Salt Tolerance in RILs Derived from SR86 × 9311

Salt-tolerant plants were randomly selected from sixth generation RILs derived from SR86 and 9311 hybrids, treated with 9‰ NaCl for 4 days at the three-leaf stage, and then allowed to recover for 4 days (Figure 5A–C). The survival rates after recovery from salt stress treatment were determined. The survival rate of the parental line 9311 was approximately 10%, and that of SR86, the other parental line, was more than 80%. The survival rates of the salt-tolerant hybrids were between 60–70% (Figure 5D). The genotypes of the RILs were verified using specific marker primers based on differences between the SR86 and 9311 sequences. Agarose gel electrophoresis showed that plants with higher salt tolerance were genetically more similar to SR86, while individuals with poorer salt tolerance resembled 9311 genetically (Figure 5E). These results confirmed that the molecular markers developed in this study to identify salt tolerance had high specificity.

### 2.6. Analysis of the Interaction between OsJRL45 and OsJRL40

Structural analysis of *OsJRL45* and *OsJRL40* revealed that both genes were located on chromosome 4 (Figure 2A). To determine whether the OsJRL45 and OsJRL40 proteins interacted, OsJRL45 was used as bait in yeast two-hybrid (Y2H) experiments. The yeast strain Y2HGold was co-transformed with the vectors *AD-OsJRL45* and *BD-OsJRL40*. Co-transformed yeast grew normally on SD/-Trp/-Leu/-His/-Ade + X-α-Gal medium (Figure 6A), indicating an interaction between OsJRL45 and OsJRL40.

To verify this Y2H result, bimolecular fluorescence complementation (BiFC) assays were conducted in tobacco leaves. Previous studies showed that OsJRL45 and OsJRL40 are located in the endoplasmic reticulum and cytoplasm, respectively [33,34]. Following co-expression of OsJRL40-YFP^N^/OsJRL45-YFP^C^, fluorescent signals were detected in the nucleus and cytoplasm (Figure 6B). Taken together, the Y2H and BiFC results indicated that there was an interaction between OsJRL45 and OsJRL40; we therefore speculated that co-expression of *OsJRL45* and *OsJRL40* in rice might enhance their function.

### 2.7. Analysis of Agronomic Traits in Mature Transgenic Plants Harboring OsJRL45/40 from SR86

To determine whether the phenotype and agronomic traits associated with SR86 could be observed in transgenic Nip plants harboring *OsJRL45* and *OsJRL40* (*OsJRL45/40*) from SR86, the T_2_ generation of transgenic plants was observed at maturity to compare their traits and phenotype with wild-type (WT) Nip (Figure 7). The single spike structure and overall phenotype of transgenic plants did not differ significantly from that of WT (Figure 7A,B); in addition, there were no significant differences in grain length, grain width, plant height, effective panicle, and seed-setting rate between transgenic plants and WT plants (Figure 7C–F). There were, however, significant differences between transgenic and WT plants in thousand-grain weight, spike length, and the number of grains per spike (Figure 7G–I). These results indicated that, although co-transformation of Nip with *OsJRL45/40* from SR86 enhanced yield, it did not affect the general phenotype of the plant.

### 2.8. Co-Transformation with OsJRL45/40 from SR86 Enhances Tolerance of Salt Stress at the Three-Leaf Stage

To further verify whether co-transforming Nip with *OsJRL45* and *OsJRL40* from SR86 enhanced tolerance of salt stress, WT and transgenic plants at the three-leaf stage were subjected to 9‰ NaCl salt stress treatment for 4 days and allowed to recover for 4 days before their phenotypes were observed (Figure 8A–C). The survival rate of the transgenic plants was significantly higher than that of WT (Figure 8D), indicating that salt tolerance was enhanced by transformation with *OsJRL45/40*. Next, the expression of *OsJRL45* and *OsJRL40* in plants exposed to salt stress for 3, 6, 9, or 12 h was quantified using RT-qPCR. This analysis found that expression levels of these genes were significantly higher in the transgenic plants than in the WT (Figure 8E,F). These results confirmed that, at the three-leaf stage, co-transformation with *OsJRL45* and *OsJRL40* from SR86 enhanced tolerance to salt stress.

### 2.9. Co-Transformation with OsJRL45/40 Derived from SR86 Enhances Antioxidant Enzyme Activities under Salt Stress

To analyze the effects of co-transforming Nip WT with *OsJRL45* and *OsJRL40* from SR86 on physiological parameters related to ROS under salt stress, plants at the three-leaf stage were treated with 9‰ NaCl for 24 h. The enzymatic activities of CAT, POD, and SOD were measured in WT and *OsJRL45/40* transgenic plants before and after salt stress treatment, as were the levels of malondialdehyde (MDA) and proline (Pro). The activities of CAT, POD, and SOD, as well as Pro content, were significantly higher in *OsJRL45/40* transgenic plants than in WT, while the MDA content showed the opposite result (Figure 9A–E). This showed that co-transforming Nip with *OsJRL45/40* from SR86 enhanced salt stress tolerance, and improved the homeostasis of related physiological parameters, including enzyme activities.

### 2.10. Analysis of Differentially Expressed Genes

To investigate the salt tolerance of transgenic Nip plants harboring *OsJRL45* and *OsJRL40* from SR86 further, we analyzed differential gene expression before and after salt stress treatment, using Fold Change ≥ 2 and q-value < 0.05 as criteria for determining differentially expressed genes (DEGs). The overlaps between groups of DEGs identified in transgenic and WT plants before and after salt stress treatment are shown in Figure 10A. We identified 1044 DEGs (638 up-regulated and 406 down-regulated) in OsJRL45/40-vs-WT before salt stress treatment and identified 834 DEGs (631 up-regulated and 203 down-regulated) in OsJRL45/40-vs-WT after salt treatment (Figure 10A); this group may contain key genes governing the response to salt stress. Kyoto Encyclopedia of Genes and Genomes (KEGG) enrichment analysis showed that genes that were differentially expressed between WT and transgenic plants before salt stress treatment were involved in “mitogen-activated protein kinase (MAPK) signaling pathway”, “plant hormone signal transduction”, and “phenylpropanoid biosynthesis” (Figure 10B). Gene ontology (GO) enrichment analysis showed that the DEGs were mainly involved in “metabolic process”, “response to stress”, “catalytic activity”, and “response to abiotic stimulus” (Figure 10C). 

Volcano plot analysis found that 631 genes were significantly up-regulated and 203 genes significantly down-regulated in transgenic plants, relative to WT, after salt stress treatment (Figure 11A,B). KEGG enrichment analysis showed these DEGs were involved in “linoleic acid metabolism”, “photosynthesis”, “alpha-linolenic acid metabolism”, “2glyoxylate and dicarboxylate metabolism”, “carbon fixation in photosynthetic organisms”, and “alanine, aspartate, and glutamate metabolism” pathways (Figure 11C). GO enrichment analysis revealed that most of these DEGs were involved in “metabolic process”, “response to stress”, and “response to abiotic stimulus” (Figure 11D). That a large number of genes were differentially expressed in a short period soon after salt stress treatment indicated that the response to salt stress occurred more rapidly in transgenic plants harboring *OsJRL45/40* from SR86 than in WT plants.

### 2.11. Gene Set Enrichment Analysis (GSEA) of the Response to Salt Stress

Further analysis of the Biological Process (BP) using GSEA revealed 104 DEGs were in the process “response to stress”, and 65 DEGs were in the process “response to abiotic stimulus” (Figure 12 and Figure 13). Both of these processes include Na^+^ transport proteins (Figure 10 and Figure 11), such as sodium transport protein (*OsHKT1;5*), low-affinity sodium transport protein (*OsHKT1;3*), and high-affinity sodium transport protein (*OsHKT2;1*). When rice plants are under salt stress, these transport proteins transport excess Na^+^ from the shoots to the roots, preventing excessive Na^+^ accumulation in the shoots, thereby reducing Na^+^ toxicity and enhancing salt tolerance. These processes also include lipoxygenase genes (*OsLOX1*, *OsHI-LOX*, *OsLOX8*, and *OsLOX11*; Figure 12 and Figure 13). Additionally, two key enzymes in the jasmonic acid (JA) biosynthesis pathway, lipoxygenase (*OsLOX2*) and allene oxide synthase (*OsAOS2*), are in the “response to stress” process (Figure 12). The JA biosynthesis pathway is involved in various signal transduction pathways and regulates physiological and molecular processes in plants; it plays an important signaling role in plant stress responses and thereby in protecting plants from salt stress. Finally, these processes both include E3 ubiquitin ligase (*OsBTBZ1*), galactinol synthase (*OsGolS1/2*), and stress membrane protein (*OsSMP1*), which are all involved in the regulating of the abscisic acid signaling pathway in response to salt stress. Glutamine synthetase (*OsGS2*) is involved in regulating the response to non-ionic osmotic stress, improving salt stress tolerance. Disease resistance protein (*OsRSR1*) is involved in ROS metabolism and plays an important role in maintaining the integrity of the biological membrane system and in defense against membrane lipid peroxidation under salt stress.

A bZIP transcription factor (*OsbZIP48*) and an MYB transcription factor (*Osmyb2*) were found in the “response to abiotic stimulus” process (Figure 13). bZIP transcription factors are involved in the biosynthesis of hormones such as gibberellin, JA, and auxin, while MYB transcription factors regulate gene expression, including of their own genes, by directly binding DNA. All of these results indicated that *OsJRL45/40* co-transgenic plants could respond more quickly to salt stress through a variety of biological processes, including “Na^+^ transport protein”, “plant hormone regulation pathways”, “non-ionic osmotic stress”, “reactive oxygen species metabolism”, and “DNA transcription factors”.

## 3. Discussion

### 3.1. OsJRL45 and OsJRL40 Interact Together to Enhance the Salt Tolerance of Plants

Salt stress has a significant impact on plant growth and development, inhibiting seed germination, root and shoot growth, and plant morphology [40]. Many rice varieties are salt-sensitive and severely affected by salt stress. Lectins have been identified in various plant species [41,42,43,44,45], but their roles in responses to stress stimuli are not yet fully understood. We previously demonstrated that two genes encoding lectins, *OsJRL45* and *OsJRL40*, regulate and enhance tolerance to salt stress in rice [34,35]. Since these genes are located close together on chromosome 4, understanding whether they interact and jointly enhance salt tolerance is of great significance.

Salt stress is a common abiotic stress that adversely affects plant development, growth, and production [33,46]. The responses of three rice cultivars, SR86, 9311, and Nip, to salt stress treatment indicated that *OsJRL45* and *OsJRL40* were associated with enhanced tolerance of salt stress in SR86 (Figure 1). Yeast two-hybrid and BiFC assays showed that the OsJRL45 and OsJRL40 proteins interacted (Figure 6). Transforming Nip with *OsJRL45* and *OsJRL40* (*OsJRL45/40*) genes from SR86 did not affect the plant phenotype (Figure 7), although salt tolerance was enhanced in transgenic plants at the three-leaf stage (Figure 8). In rice, salt stress disrupts the balance between reactive oxygen species (ROS) production and scavenging, leading to excessive ROS accumulation, which disrupts plant physiological functions and impedes normal growth and development [47,48]. CAT, POD, and SOD activities increased significantly in the transgenic plants, and their MDA content was significantly lower than that of the WT (Figure 9). Previous studies showed that *OsJRL45* and *OsJRL40* enhance the antioxidant capacity of cells and prevent excessive ROS accumulation [34,35]. The results of transformation with *OsJRL45/40* from SR86 were consistent with these observations, indicating that *OsJRL45* and *OsJRL40* interacted and enhanced salt stress tolerance. High salt concentrations damage rice root development, accelerate leaf curling and yellowing, reduce the number of grains per panicle, and decrease the thousand-grain weight, resulting in yield reduction [49,50]. Therefore, understanding the molecular mechanisms underlying salt tolerance and screening for this trait are key steps for the development of salt-tolerant rice varieties.

### 3.2. Transcriptome Analysis of OsJRL45 and OsJRL40 Transgenic Seedlings under Salt Stress

Transgenic rice plants harboring *OsJRL45/40* from SR86 exhibited positive physiological responses under salt stress. Key salt-responsive metabolic pathways and DEGs were explored and analyzed using transcriptomic data. RNA-seq analysis identified 1044 DEGs (638 up-regulated and 406 down-regulated) in OsJRL45/40-vs-WT before salt stress treatment and identified 834 DEGs in OsJRL45/40-vs-WT after salt stress treatment; 631 genes were up-regulated and 203 down-regulated, compared with WT plants (Figure 10A). The results of Volcano plot analyses were consistent (Figure 11A,B). Before salt stress treatment, KEGG enrichment analysis showed that the differentially expressed genes between WT and transgenic plants were involved in the “mitogen-activated protein kinase (MAPK) signaling pathway”, “plant hormone signaling”, and “phenylpropanoid biosynthesis” (Figure 10B). GO enrichment analysis showed that DEGs were mainly involved in “metabolic processes”, “responses to stress”, “catalytic activity”, and “responses to abiotic stimulus” (Figure 10C). KEGG enrichment analysis after salt stress treatment revealed that these DEGs were involved in “linoleic acid metabolism”, “photosynthesis”, “alpha-linolenic acid metabolism”, “2-glyoxylate and dicarboxylate metabolism”, “carbon fixation in photosynthetic organisms”, and “alanine, aspartic acid, and glutamate metabolism” pathways (Figure 11C). GO enrichment analysis revealed that most of these DEGs were involved in “metabolic processes”, “responses to stress”, and “responses to abiotic stimulus” (Figure 11D). Soon after salt stress treatment, a large number of genes were differentially expressed in a short period of time, indicating that the transgenic plants of *OsJRL45/40* of SR86 could respond more quickly to salt stress than WT plants. GO functional analysis indicated that most DEGs were involved in the pathways “response to stress” and “response to abiotic stimulus” (Figure 10C and Figure 11D). 

GSEA of GO enrichment analysis in the biological process (BP) categories “response to stress” and “response to abiotic stimulus” (Figure 12 and Figure 13) identified the Na^+^ transporter genes *OsHKT1;5*, *OsHKT1;3*, and *OsHKT2;1*. These transporters transfer excess Na^+^ from the shoots, thus reducing Na^+^ toxicity and enhancing salt tolerance in rice under salt stress [51,52,53]. The analysis also identified four lipoxygenase genes, *OsLOX1*, *OsHI-LOX*, *OsLOX8*, and *OsLOX11*. Two key enzymes lipoxygenase (OsLOX2) and allene oxide synthase (OsAOS2), which encode key enzymes in the JA synthesis pathway, which is involved in various signaling pathways and regulates many physiological and molecular processes in plants [54,55,56,57], thus protecting plants from salt stress. Additionally, the analysis identified genes involved in regulating the ABA signaling pathway, including E3 ubiquitin ligase (*OsBTBZ1*), galactinol synthase (*OsGolS1/2*), and stress membrane protein (*OsSMP1*); such genes respond to salt stress via ABA-dependent pathways, enhancing tolerance of abiotic stress [58,59,60]. Glutamine synthetase (*OsGS2*), which is involved in regulating the response to non-ionic osmotic stress, enhances rice tolerance to salt stress. The disease resistance protein (*OsRSR1*), which is involved in the regulation of reactive oxygen metabolism, can defend against lipid peroxidation under salt stress [61], and thus plays an important role in protecting the integrity of the biological membrane structure. The analysis also identified genes involved in the “response to abiotic stimulus” pathway, including a bZIP transcription factor (*OsbZIP48*), which is involved in biosynthesis of the plant hormones gibberellin, JA, and auxin [62], and an MYB transcription factor (*Osmyb2*) that positively regulates *OsHKT1;5* to control its transcription [63]. Consistent with our previous study, co-transformation with *OsJRL45/40* enhanced antioxidant enzyme activity under salt stress through non-ionic osmotic stress and reactive oxygen metabolism processes and regulated Na^+^-K^+^ homeostasis through Na^+^ transport proteins [34,35].

### 3.3. Sequence Differences between OsJRL45 and OsJRL40: Marker-Assisted Selection (MAS) Analysis for the Selection of Salt-Tolerant Rice

The development of molecular markers that can be used to screen for salt-tolerant rice varieties is crucial for selecting plants with high salt tolerance, as salt stress significantly affects rice yield and quality [64]. Molecular marker-assisted identification and selection of genes that control resistance to these factors have been used for rice improvement [22]. Previous studies showed that *OsJRL45* and *OsJRL40* are located on chromosome 4 [34,35,39]. The *OsJRL45* and *OsJRL40* gene sequences of SR86, 9311, L623, R498, Minghui63, Zhenshan97, Yu Chi, Nantehao, Qingzhen, and Meizhan were amplified by sequencing and aligned, which revealed eight single base differences in exon 1 in the *OsJRL45* gene between these cultivars. The differences at positions 63 (base-pairs (bp)), 72 bp, and 123 bp corresponded to synonymous mutations after the initial ATG, while those at 209 bp, 224 bp, 245 bp, 263 bp, and 268 bp in exon 1 would result in amino acids changes (Figure 2). The single base change in exon 2 at position 1000 bp in the cultivars SR86 and Nantehao are A, resulting in a change to the amino acid Glu, whereas those in remaining cultivars are C, resulting in a change to amino acid Asp (Figure 2). In exon 1 of the *OsJRL40* gene (position 131 bp), only Meizhan showed a single base change to A, and the nucleotides in the remaining cultivars were C. In exon 2 (position 1123 bp), only SR86 had a single base change to A, resulting in a change to the amino acid Ser, whereas in the remainder, the nucleotide was G, resulting in the amino acid Gly (Figure 2). However, sequencing analysis of the sequences between the *OsJRL45* and *OsJRL40* genes of SR86, R498, Minghui63, and Zhenshan97 showed that the distance between the two genes was only 2605 bp, and at 204 bp downstream of *OsJRL45*, the SR86 sequence was missing the 25 bp sequence GCGCACGACGCGGCCGCCAGCGTTC, which was located 2401 bp upstream of *OsJRL40*. Based on the differences between the genes and the binding of *OsJRL45* and *OsJRL40*, molecular marker primers were designed for use as molecular markers and for validation.

The analyses of the recombinant inbred lines constructed by the hybridization of SR86 and 9311 under salt stress and the molecular marker primers developed in this study showed that the molecular-tagged primers based on the nucleotide differences between *OsJRL45* and *OsJRL40* could be used to screen salt-tolerant plants (Figure 2, Figure 3 and Appendix A and Table 1). In order to further verify the accuracy of the molecular markers developed in this study for screening salt-tolerant rice, some rice plants commonly used in production were used for analysis. Rice varieties with higher salt tolerance (Foer B, Chaimi, Yuejing, Xiangzao 32, Nantehao, R288, Shuhui 527, Aizinuo, and Xiang 628S) produced gel electrophoresis results that were very similar to those of SR86, while those of salt-intolerant plants were similar to those of 9311 (Figure 4). This suggests that the novel molecular markers designed to identify salt-tolerant plants in this study identified more than 90% of the plants with salt tolerance, indicating that they would have high specificity for the screening of salt-tolerant plants. Salt-tolerant plants were randomly selected in the sixth generation of RILs derived from the R86 and 9311 hybrids, and the genotypes of the RILs were verified using specific marker primers based on differences between SR86 and 9311 sequences. Agarose gel electrophoresis showed that plants with higher salt tolerance were more genetically similar to SR86, while individuals with less salt tolerance were genetically more similar to 9311 (Figure 5E). These results confirm the high specificity of the molecular markers developed in this study for the identification of salt-tolerant rice cultivars. This targeted screening approach minimized the number of undesired traits and reduced the risk of linkage drag while maintaining the desirable genetic background of the parent varieties. This study demonstrated the effectiveness of these novel molecular markers in screens to identify salt-tolerant rice plants, showing that molecular markers based on differences in intergenic nucleotide sequences can facilitate the process of selective breeding.

## 4. Materials and Methods

### 4.1. Experimental Plants

The WT rice varieties SR86, Nip, and 9311 used in this study were provided by the rice germplasm resource platform of the Hunan Rice Research Institute [34,35,39,65]. All rice varieties were grown in the field at the Hunan Normal University in Changsha, Hunan Province. All of the experiments reported in this paper comply with the institutional, national, and international guidelines concerning plant genetic resources.

To construct the population of SR86/9311 RILs, SR86 and 9311 plants were crossed. Self-pollinated F_1_ plants were used to produce F_2_ seeds; after harvest, isolated F_2_ progeny were self-pollinated, and individual salt-tolerant and salt-intolerant plants were selected for further inbreeding under salt stress treatment. The RIL population was constructed by continuous inbreeding under salt stress treatment for five generations. 

The *OsJRL45* and *OsJRL40* genes from SR86 were amplified by PCR and inserted into the vector pCAMBIA1300 using the gene recombination method. The relevant constructed vector primers are listed in Appendix A. Nip plants were transformed using *Agrobacterium tumefaciens*-mediated transformation to obtain *OsJRL45/40* transgenic plants. The phenotype and agronomic traits of the transgenic plants were analyzed in the T_2_ generation. In addition, a hybrid population was constructed using SR86 and 9311 as parents, from which salt-tolerant plants were screened for subsequent research.

### 4.2. Construction of Segregating Populations

SR86 and 9311 were crossed in the field at Hunan Normal University. The F_1_ generation was self-pollinated to obtain F_2_ seeds. F_2_ plants were treated with 9‰ salt stress at the seedling stage.

### 4.3. Marker-Assisted Selection

Total genomic DNA was extracted from rice leaves using the cetyltrimethylammonium bromide (CTAB) method [66]. DNA was analyzed by PCR with a total reaction volume of 10 μL in a PCR instrument (Bio-Rad, T100^TM^ Thermal Cycler, Foster City, CA, USA). The primers are listed in Appendix A. The PCR amplification regime was as follows: initial denaturation at 94 °C for 5 min, followed by 35 cycles of 94 °C for 30 s, 58 °C for 30 s, and 72 °C for 45 s, with a final extension at 72 °C for 5 min. The PCR products were separated by electrophoresis on 2% agarose gels and visualized under UV light.

### 4.4. Salt Stress Treatment of Transgenic Plants

Uniform, plump rice seeds were selected and surface-sterilized in 0.3% sodium hypochlorite for 20 min. Seeds were rinsed three times in distilled water, evenly spread across two layers of moist filter paper in a Petri dish, and incubated at 37 °C. After germination, seedlings were transplanted into a 96-well tray containing culture solution. The nutrient solution was changed every 4 days. Rice seedlings were grown under conventional hydroponic conditions until the three-leaf stage (approximately 14 days), then transferred to a hydroponic solution containing 9‰ NaCl for 4 days, and finally moved to a conventional hydroponic solution for recovery for 4 days.

### 4.5. Measurement of Physiological Parameters

Superoxide dismutase (SOD), peroxidase (POD), catalase (CAT), malondialdehyde (MDA), and proline contents were measured according to the kit manufacturer’s instructions (Jiancheng Bioengineering Institute, Nanjing, China) [34,35,39]. Measurements were made before treatment and 24 h after salt stress treatment. All experiments involved three independent biological replicates; data are reported as the mean ± standard deviation (SD).

### 4.6. RNA Extraction and Quantitative Real-Time PCR (qRT-PCR)

Rice seedlings at the three-leaf stage from the WT and *OsJRL45/40* transgenic lines were treated with 9‰ NaCl. Total RNA was extracted using the TRIzol Reagent (Invitrogen, Waltham, MA, USA) and reverse transcribed using the HiScript II Q RT SuperMix for qPCR (+gDNA wiper) (Vazyme, R223-01, Nanjing, China). qRT-PCR was performed using the ChamQ Universal SYBR qPCR Master Mix (Vazyme, Q711-02, Nanjing, China), with rice *Osactin* as the internal reference gene. Relative gene expression was calculated using the 2^−ΔΔCt^ method and expressed as the mean ± SD [67]. All primers used in the RT-qPCR analysis are listed in the Appendix A.

### 4.7. Transcriptomic Data Analysis

Whole WT and *OsJRL45/40* transgenic plants were sampled at the three-leaf stage, before treatment, and 24 h after 9‰ NaCl stress treatment. Three biological replicates were collected for each plant. Samples were frozen in liquid nitrogen and sent on dry ice to OE Biotech Co. Ltd., Shanghai, China for transcriptomic analysis. The RNA-seq data were analyzed as described previously [34,35,68]. The bioinformatics analysis and graphical plotting were performed using the OE Biotech cloud platform. Bioinformatic analysis was performed using the OE Cloud tools at https://cloud.oebiotech.com/task/ accessed on 21 December 2023. The volcano map and other graphics were drawn using R (https://www.r-project.org/, accessed on 25 December 2023) on the OE Cloud platform (https://cloud.oebiotech.com/task/, accessed on 25 December 2023).

### 4.8. Yeast Two-Hybrid (Y2H) Assays

Yeast two-hybrid assays were performed using the GAL4 yeast two-hybrid system. The coding sequence (CDS) of *OsJRL45* was cloned into the pGADT7 vector to generate *AD-OsJRL45*. The CDS of *OsJRL40* was cloned into the pGBKT7 vector to generate *BD-OsJRL40*. The relevant primers are listed in the Appendix A. The Y2HGold yeast strain was co-transformed with the *AD-OsJRL45* and *BD-OsJRL40* constructs and grown on the selection media SD/-Trp/-Leu (lacking tryptophan and leucine) and SD/-Trp/-Leu/-His/-Ade + X-α-Gal (lacking adenine, tryptophan, leucine, and histidine) [69].

### 4.9. Bimolecular Fluorescence Complementation (BiFC) Assay

The CDS of *OsJRL45* was fused to the C-terminus of yellow fluorescent protein (YFP), and the CDS of *OsJRL40* was fused to the N-terminus of YFP to construct the co-expression vectors OsJRL40-YFP^N^ and OsJRL45-YFP^C^. The specific primers used are listed in the Appendix A. The plasmids were introduced into *Agrobacterium tumefaciens* (GV3101) by electroporation, and the bacterial suspension was adjusted to an OD 600 of approximately 0.6. Tobacco plants (*Nicotiana benthamiana*) with good growth were selected. The lower epidermis of their leaves was injected with the bacterial suspension and labeled [70]. Tobacco plants were cultured under low light for 2 days following injection. The injected leaves were sampled, mounted on slides, and observed under a confocal laser scanning microscope (Nikon C2-ER) for image capture and recording.

### 4.10. Statistical Analysis

Data are presented as the mean ± SD of three or more independent replicates. The mean of the replicates for each experiment was calculated using PASW Statistics 18 software. Statistical significance was determined using one-way ANOVA and Student’s *t*-test with *p* < 0.05. Bar charts were created using GraphPad Prism 9. Heat maps were constructed and drawn using TBtools-II v.2.119 software [71].

## 5. Conclusions

Based on the results of the treatments and RT-qPCR analysis of salt-tolerant cultivars SR86, 9311, and Nip under salt stress, we speculated that *OsJRL45* and *OsJRL40* genes contributed to the stronger salt tolerance of SR86. To further validate this hypothesis, we analyzed the sequence differences between the *OsJRL45* and *OsJRL40* genes in SR86 and 9311 using sequencing technology, and developed molecular markers for salt-tolerant screening based on these differences. Subsequently, we constructed a recombinant inbred line population using salt-tolerant SR86 and conventional variety 9311 as parental varieties and verified the effectiveness of the developed molecular markers using rice varieties commonly used for rice production.

Yeast two-hybrid and bimolecular fluorescence complementation further demonstrated the interaction between OsJRL45 and OsJRL40 proteins. In addition, *OsJRL45/40* co-transgenic plants were created by genetic recombination technology, and salt stress treatment and physiological parameters were measured. The results showed that compared with the wild type, the antioxidant enzyme activities (including CAT, POD, SOD) and proline (Pro) content of *OsJRL45/40* co-transgenic rice were significantly increased, while their malondialdehyde (MDA) content was significantly reduced, suggesting that synergy between *OsJRL45* and *OsJRL40* increased salt tolerance.

In addition, transcriptome analysis revealed 834 differentially expressed genes (DEGs) in the *OsJRL45/40* co-transgenic plants under salt stress conditions. GO and KEGG enrichment analysis showed that these differential genes were mainly involved in metabolic pathways related to antioxidant response and osmotic homeostasis, which are known to play a key role in salt stress tolerance.

In summary, synergy between the activities of *OsJRL45* and *OsJRL40* not only increased salt tolerance in rice, but the development of molecular markers based on differences in the sequences of *OsJRL45* and *OsJRL40* between the salt-tolerant variety SR86 and the salt-sensitive variety 9311 was successful in identifying salt-tolerant rice varieties.

## Figures and Tables

**Figure 1 ijms-25-10912-f001:**
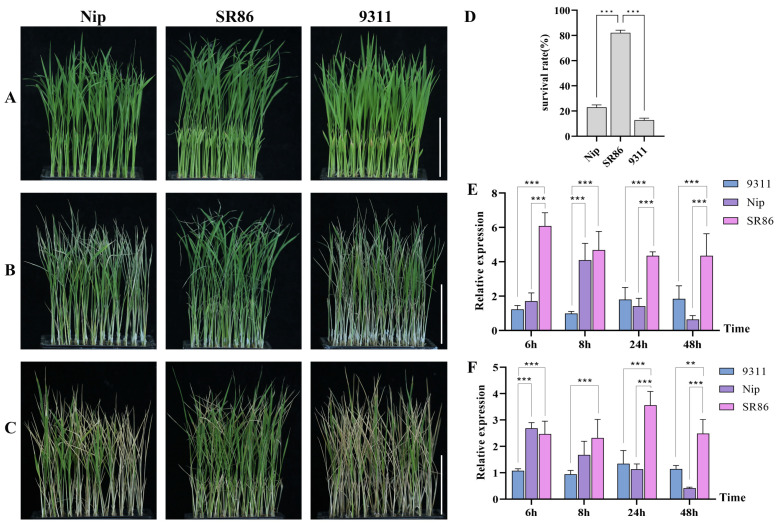
Plant phenotypes and analysis of *OsJRL45* and *OsJRL40* expression at the three-leaf stage under 9‰ NaCl salt stress. (**A**) Appearance of plants at the three-leaf stage before salt stress treatment. (**B**) Appearance of plants after 4 days of treatment with 9‰ NaCl. (**C**) Appearance of plants after recovery for 4 days. (**D**) Survival rates after recovery from salt stress treatment. (**E**) Quantification of RT-qPCR analysis of *OsJRL45* expression. (**F**) Quantification of RT-qPCR analysis of *OsJRL40* expression. Data represent the mean ± SD of three independent replicates; asterisks indicate statistically significant differences (**: *p* < 0.01; ***: *p* < 0.001). Scale bars on (**A**–**C**): 5 cm.

**Figure 2 ijms-25-10912-f002:**
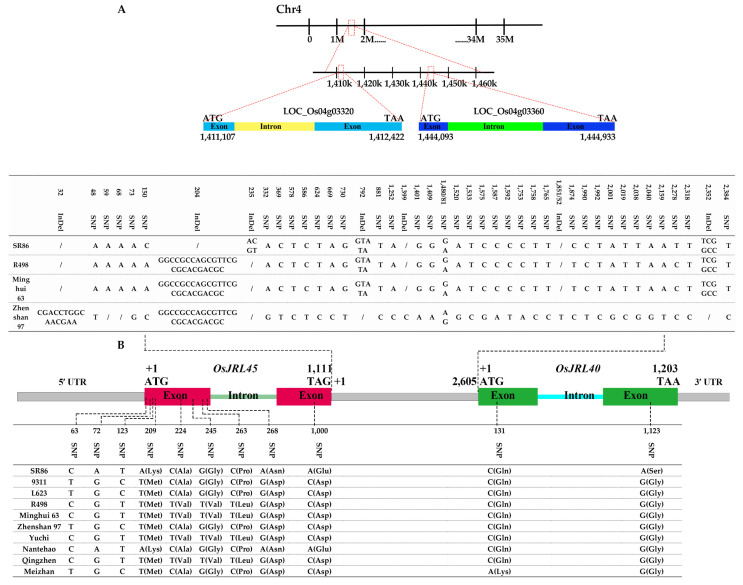
Location and structure of *OsJRL45* and *OsJRL40*. (**A**) Structure of *OsJRL45* and *OsJRL40* in Nip. ATG indicates the start codon. (**B**) Positions of single nucleotide polymorphisms (SNPs) in the exons of *OsJRL45* and *OsJRL40* and the resulting changes in amino acid sequences.

**Figure 3 ijms-25-10912-f003:**
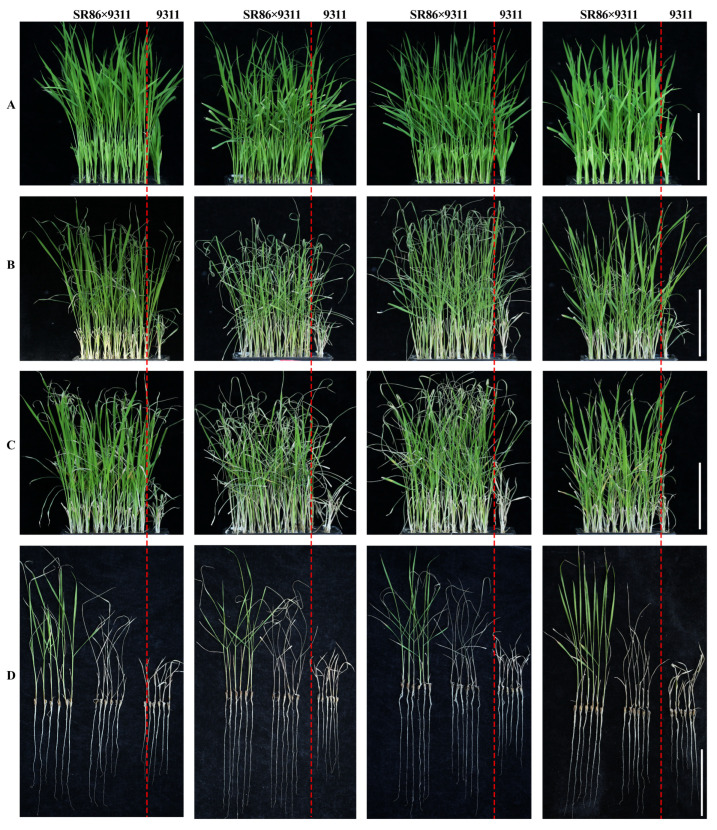
Effect of salt stress on plants from a hybrid population derived from the cross SR86 × 9311. (**A**) Appearance of plants at the three-leaf stage before salt stress treatment. (**B**) Appearance of plants after 5 days of 6‰ NaCl salt stress treatment. (**C**) Appearance of plants after 5 days of recovery. (**D**) Appearance of individual plants after recovery. SR86 × 9311 indicates plants from the hybrid population; plants from the parental line 9311 are shown to the right of the red dotted line. Scale bars on (**A**–**D**): 5 cm.

**Figure 4 ijms-25-10912-f004:**
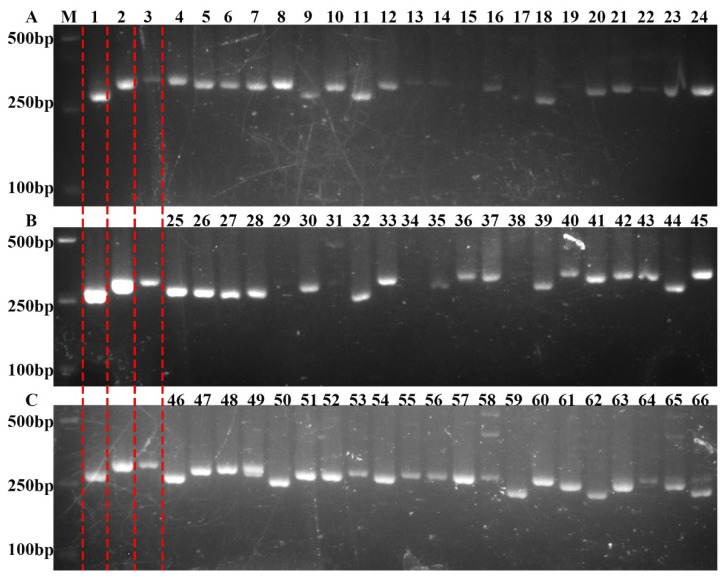
Validation of marker-assisted selection in different rice varieties. Lanes 1, 2 and 3 (indicated by the red dotted lines) represent SR86, 9311, and Nip, respectively. (**A**) From left to right (lanes 4–24): Mei Zhan, Yu Chi, Ming Hui, Zhan F2, Feng Hua Zhan, Fu Yi B, IR64, Chai Mi, Zi 100, B213, Zhu 1S, Guangxiang 24S, Hua 451, Xiang Wan Indica, Yuejing, Xianhui, bolon, Zm273 Bairizao, Zhen Indica 97A, Guangzhan 63S. (**B**) From left to right (lanes 25–45): TTP, Qingzhen, TB, Lemont, Wanhui 110, PA64S, ZhongjiuA, Xiangzao 32, Gang 46B, 7001S2, CHEN 28, Sanyingzhan, Yongping, Qingzhen, Guangsi, IR36, 9311, Nan 11, Xiangzao Indica, IR24. (**C**) From left to right (lanes 46–66): Nantehao, Ludao 2318, R402, R16, R288, Minghui 63, P002, Qiuguang, Shuhui 527, Yujing 1, Zhengxuan 3, Xiaozinuo, D16, Xiangzi 3150, Xiangxiang 628S, Huaihui 210, Yixiang A, Duozi 1, Lila 2, Yuehui A, 750S. M: DNA size marker.

**Figure 5 ijms-25-10912-f005:**
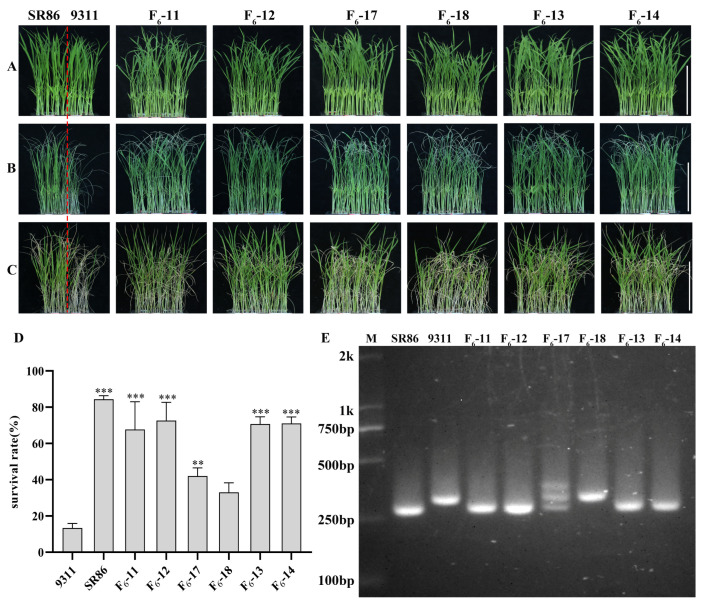
Screening of RILs derived from SR86 × 9311 for salt tolerance. (**A**) Appearance of plants at the three-leaf stage before salt stress treatment. (**B**) Appearance of plants after 4 days of treatment with 9‰ NaCl. (**C**) Appearance of plants after 4 days of recovery from salt stress. F_6_ indicates the sixth generation of RILs derived from the cross SR86 × 9311. (**D**) Survival rate statistics: data represent the mean ± SD of three independent replicates; asterisks indicate statistically significant differences (**: *p* < 0.01; ***: *p* < 0.001). (**E**) Genotyping of hybrid plants using specific molecular markers. (**A**–**C**) On the left side of the red dotted line is SR86 and on the right is 9311. M: DNA size marker. Scale bars on (**A**–**C**): 5 cm.

**Figure 6 ijms-25-10912-f006:**
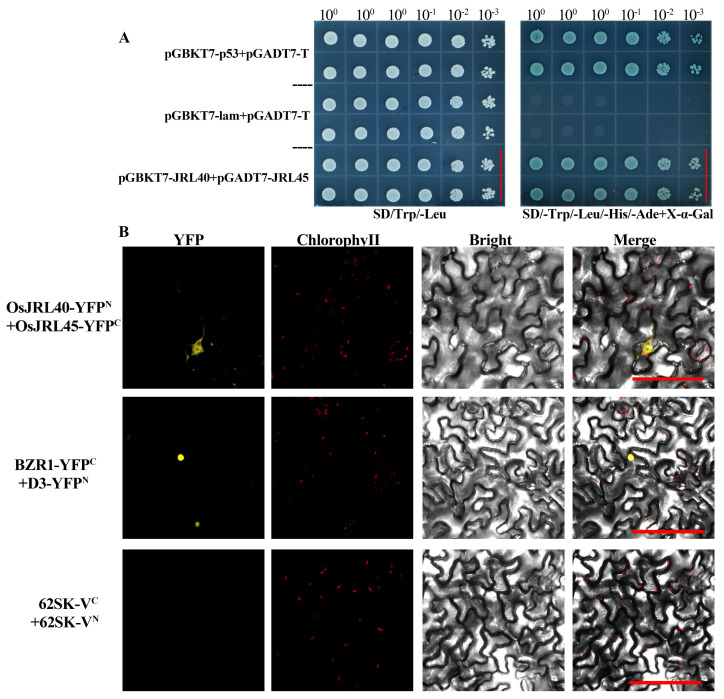
Interaction between OsJRL45 and OsJRL40. (**A**) Yeast two-hybrid (Y2H) assays showing the in vivo interaction between OsJRL45 and OsJRL40. The recombinant plasmids pGBKT7-53/pGADT7-T and pGBKT7-Lam/pGADT7-T were used as positive and negative controls, respectively. SD/-Trp/-Leu medium lacks tryptophan and leucine; SD/-Trp/-Leu/-His/-Ade + X-α-Gal medium lacks adenine, tryptophan, leucine, and histidine. (**B**) Bimolecular fluorescence complementation (BiFC) assay confirming the interaction between OsJRL45 and OsJRL40. Yellow indicates yellow protein fluorescence and red indicates chloroplast channels. BZR1-CYFP/D3-NYFP and 62SK-VC/62SK-VN were used as positive and negative controls, respectively. (**A**) Scale bars: 2 cm, (**B**) Scale bars: 20 μm.

**Figure 7 ijms-25-10912-f007:**
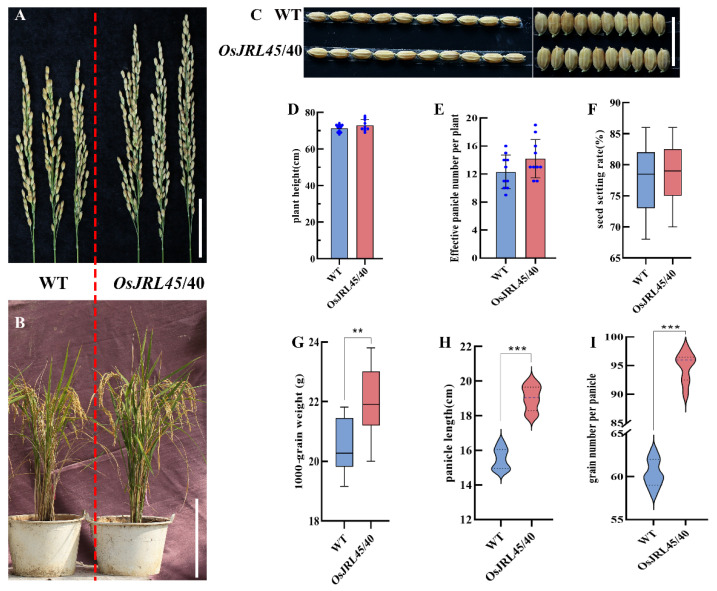
Analysis of agronomic traits in transgenic plants harboring *OsJRL45/40* and Nip wild-type (WT) rice. (**A**) Single spike phenotype. (**B**) Mature plant phenotype. (**C**) Grain length and width phenotypes. (**D**) Plant height. (**E**) Effective panicle. (**F**) Seed-setting rate. (**G**) Thousand-grain weight. (**H**) Spike length. (**I**) Number of grains per spike. Data represent the mean ± SD of three independent replicates; asterisks indicate statistically significant differences (**: *p* < 0.01; ***: *p* < 0.001). (**A**,**B**) On the left side of the red dotted line is WT and on the right is *OsJRL45/40*. Scale bars on (**A**–**C**): 5 cm.

**Figure 8 ijms-25-10912-f008:**
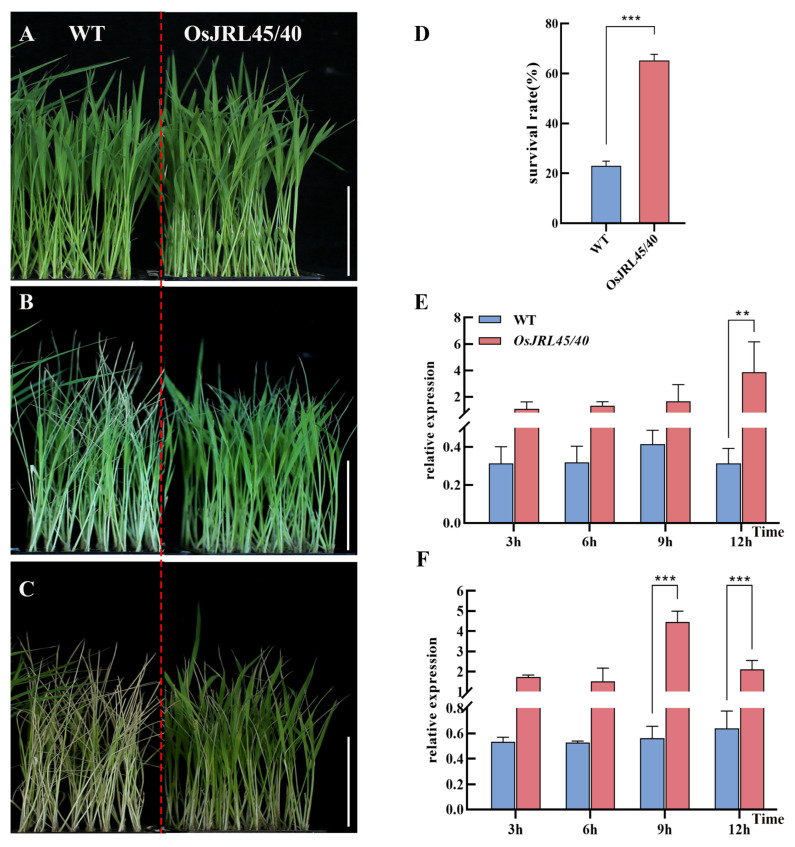
Phenotypic analysis of plants at the three-leaf stage exposed to salt stress by treatment with 9‰ NaCl. (**A**) Appearance of plants at the three-leaf stage before salt stress treatment. (**B**) Appearance of plants after 4 days of treatment with 9‰ NaCl. (**C**) Appearance of plants after 4 days of recovery from salt stress. (**D**) Survival rates after recovery from salt stress. (**E**) Quantification of *OsJRL45* expression by RT-qPCR. (**F**) Quantification of *OsJRL40* expression by RT-qPCR. WT: wild-type Nip; *OsJRL45/40*: transgenic plants. Data represent the means ± SD of three independent replicates; asterisks indicate statistically significant differences (**: *p* < 0.01; ***: *p* < 0.001). (**A**–**C**) On the left side of the red dotted line is WT and on the right is *OsJRL45/40*. Scale bars on (**A**–**C**): 5 cm.

**Figure 9 ijms-25-10912-f009:**
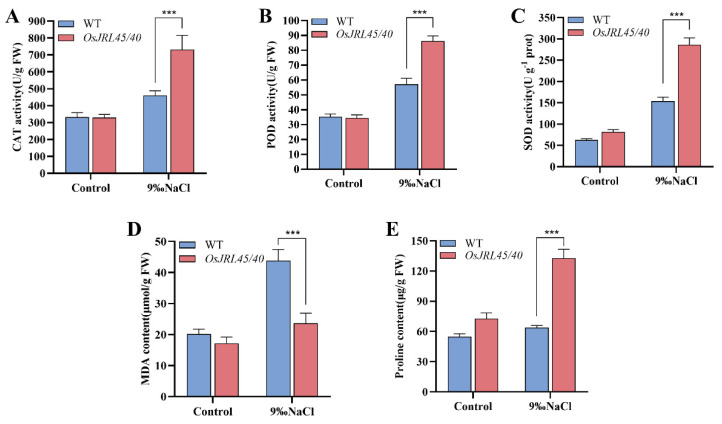
Changes in physiological indicators of plants at the three-leaf stage treated with 9‰ NaCl for 24 h. (**A**) Catalase (CAT) activity. (**B**) Peroxidase (POD) activity. (**C**) Superoxide dismutase (SOD) activity. (**D**) Malondialdehyde (MDA) content. (**E**) Proline (Pro) content. WT: wild-type Nip; *OsJRL45/40*: transgenic plants. Data represent the mean ± SD of three independent replicates; asterisks indicate statistically significant differences (***: *p* < 0.001).

**Figure 10 ijms-25-10912-f010:**
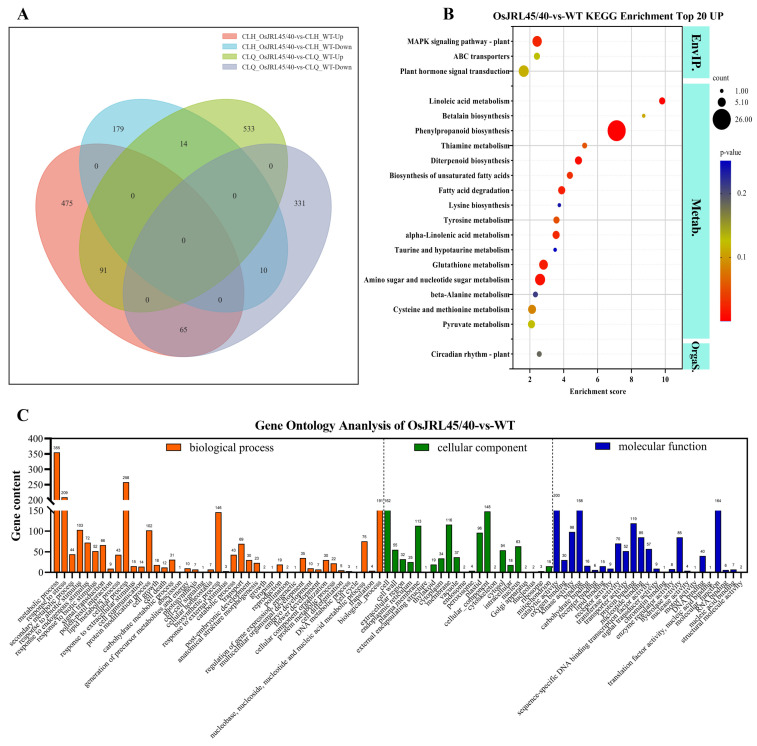
GO and KEGG enrichment analyses of differential gene expression before and after salt stress treatment. (**A**) Venn diagram showing the overlap of differentially expressed genes (DEGs) identified in WT and transgenic seedlings harboring *OsJRL45/40* from SR86 before and after salt stress treatment. CLQ-OsJRL45/40-vs-CLQ-WT-Up/Down: genes that were up-regulated and down-regulated in transgenic plants, relative to their expression in WT plants, under normal conditions; CLH-OsJRL45/40-vs-CLH-WT-Up/Down: genes that were up-regulated and down-regulated in transgenic plants, relative to their expression in WT plants, under salt stress conditions. (**B**) KEGG enrichment analysis of genes differentially expressed in WT and transgenic plants before salt stress treatment. (**C**) Functional annotation of genes differentially expressed in WT and transgenic plants before salt stress treatment.

**Figure 11 ijms-25-10912-f011:**
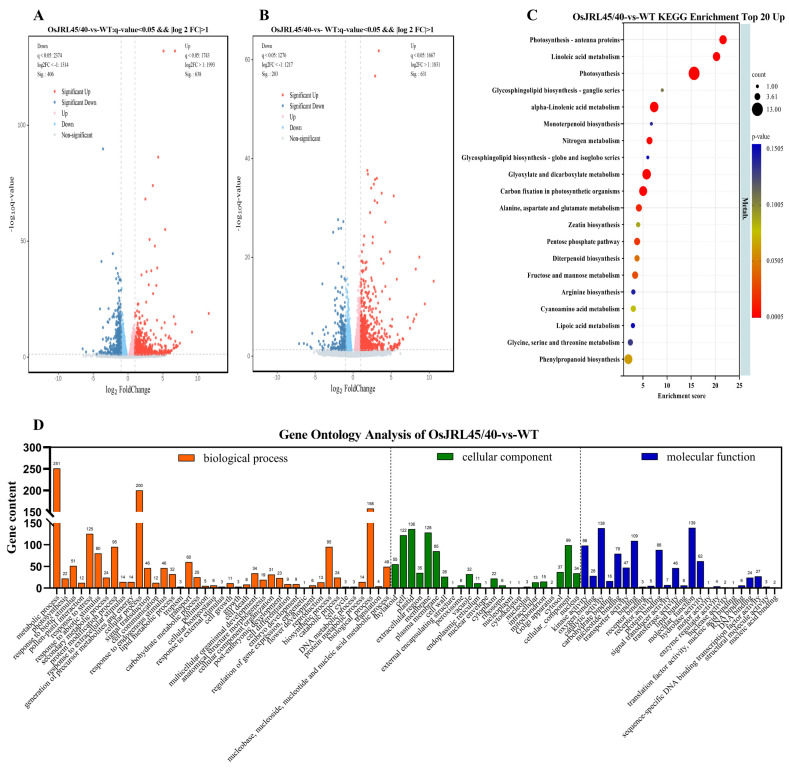
GO and KEGG enrichment analyses of DEGs detected in WT and transgenic plants harboring *OsJRL45/40* from SR86 after salt stress treatment. Volcano plot of DEGs detected (**A**) before and (**B**) after salt stress treatment. (**C**) KEGG enrichment analysis of DEGs detected after salt stress treatment. (**D**) Classification and functional annotation of DEGs between WT and transgenic plants after salt stress treatment.

**Figure 12 ijms-25-10912-f012:**
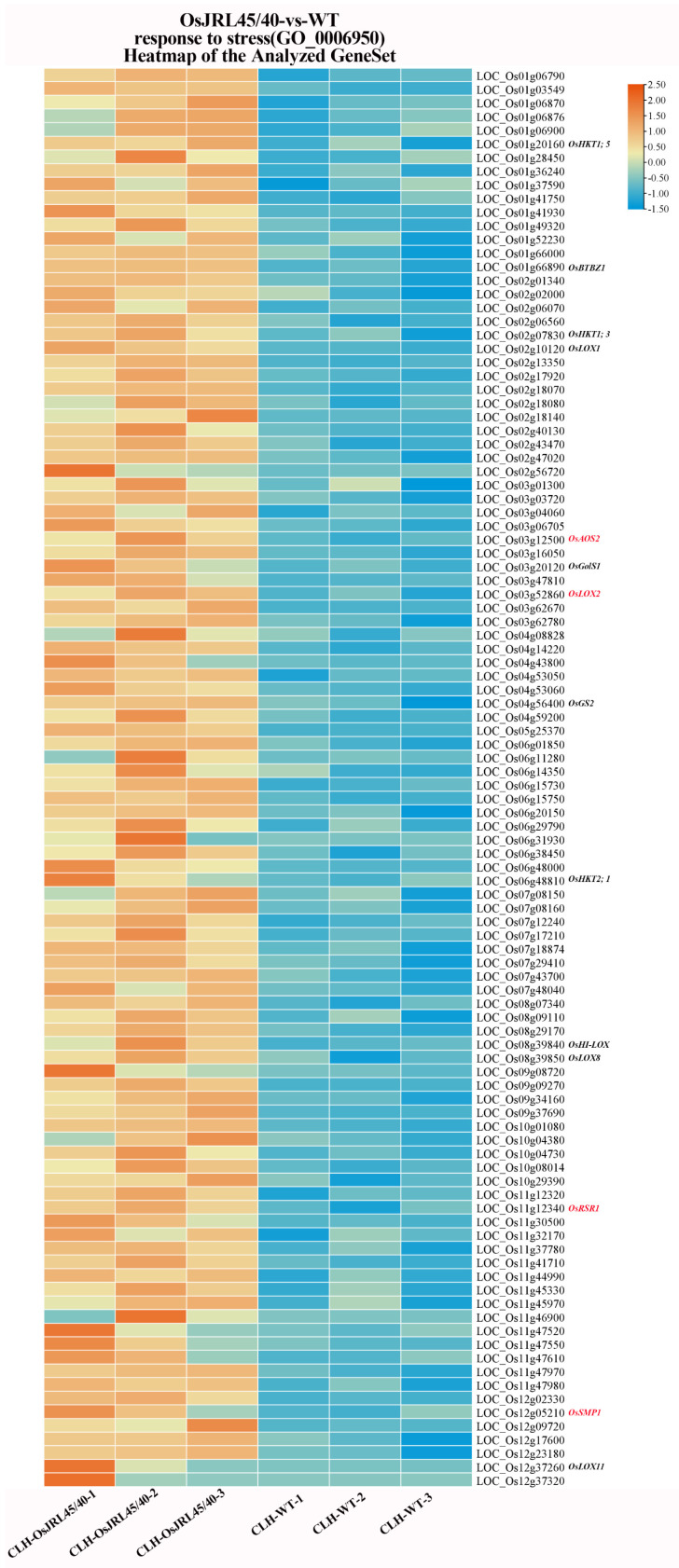
DEGs enriched in “response to stress” GO pathways after salt stress treatment, identified by GSEA. Genes that also appear in Figure 13 are shown in black; genes shown in red are unique to this process.

**Figure 13 ijms-25-10912-f013:**
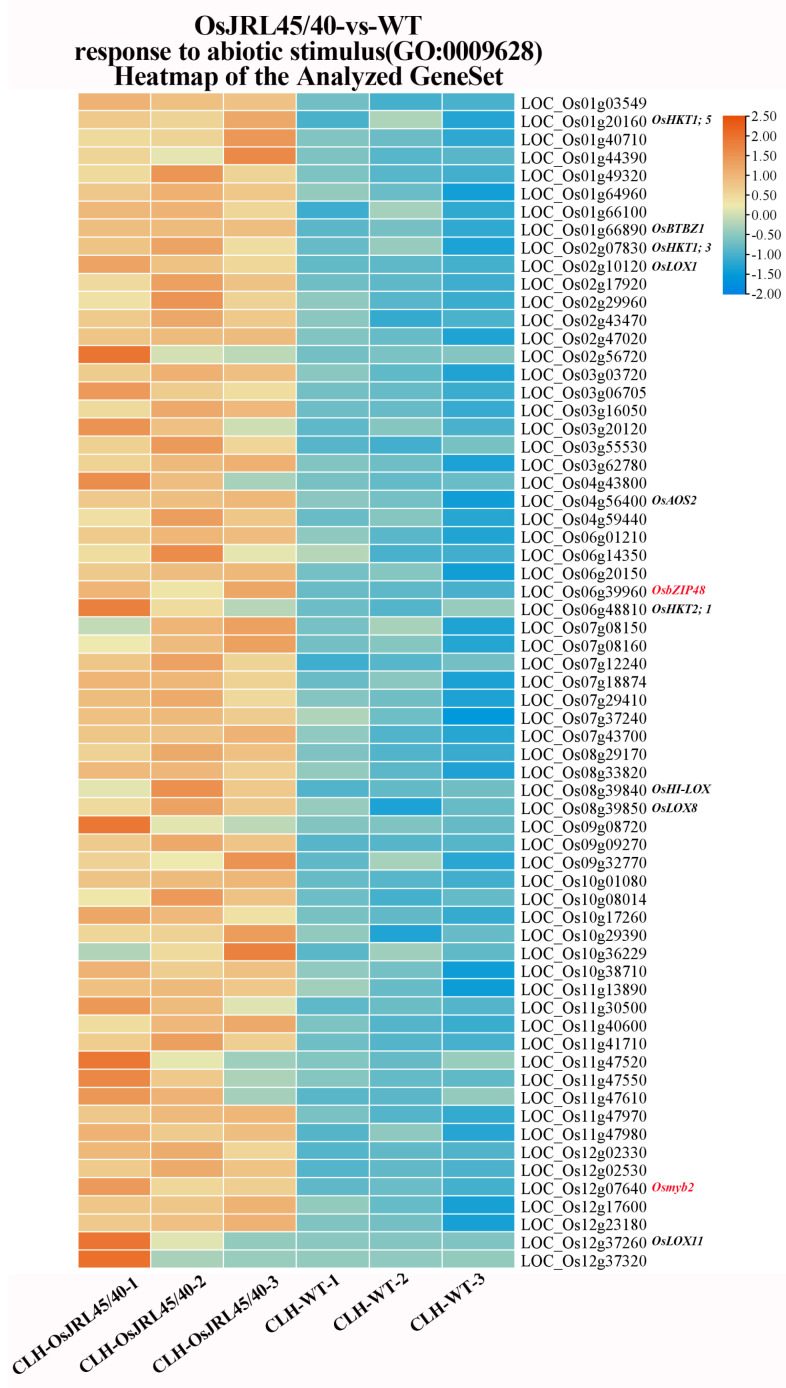
DEGs enriched in “response to abiotic stimulus” GO pathways after salt stress treatment, identified by GSEA. Genes that also appear in Figure 12 are shown in black; genes shown in red are unique to this process.

**Table 1 ijms-25-10912-t001:** Validation of MAS of salt stress in 320 SR86 × 9311 RILs.

	Number of SR86 × 9311 RILs
SR86	228
9311	54
Heterozygous	36
No product	2
Total	320

## Data Availability

All of the data supporting the conclusions of this article are provided within the article and in Appendix A. All of the data and materials are available upon reasonable request from the corresponding author.

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
