# Peer review of "Marker-Assisted Selection of Jacalin-Related Lectin Genes OsJRL45 and OsJRL40 Derived from Sea Rice 86 Enhances Salt Tolerance in Rice"

_ijms, 2024, doi:10.3390/ijms252010912_

Round 1

Reviewer 1 Report

Comments and Suggestions for Authors

The manuscript, “Marker-assisted selection of jacalin-related lectin genes osjrl45 and osjrl40 derived from sea rice 866 enhance salt tolerance in rice,” investigated the function of OsJRL45 and OsJRL40 genes in recombinant inbred lines from SR86 and 931 hybrid populations. The manuscript was well-written, and the results were clearly explained. 

I found that the affiliations needed to be checked and corrected. Additionally, some keywords exist in the title; please select different keywords that do not exist in the title. 

The introduction is long; please refrain from repeating subjects in the introduction section. 

Lastly, the authors have a lot of data and findings; the discussion was insufficient for readers. Please improve the discussion section carefully. 

Author Response

Comments 1: I found that the affiliations needed to be checked and corrected. Additionally, some keywords exist in the title; please select different keywords that do not exist in the title.

Response 1: Thank you for pointing this out. We agree with this comment. We have revised the keywords, delete keywords that duplicate the title, the revised manuscript this change can be found page number 1, line 26-27. Updated text in the manuscript if necessary. Thank you.

Comments 2: The introduction is long; please refrain from repeating subjects in the introduction section.

Response 2: Thank you for pointing this out. We agree with this comment. We have revised the repetition of the subjects in the introduction section, the revised manuscript this change can be found page number 1-3, line 29-101. Updated text in the manuscript if necessary. Thank you.

Comments 3: Lastly, the authors have a lot of data and findings; the discussion was insufficient for readers. Please improve the discussion section carefully.

Response 3: Thank you for pointing this out. We agree with this comment. We have revised the discussion section, the discussion in three points: 3.1 OsJRL45 and OsJRL40 interact together to enhance the salt tolerance of plants, 3.2 Transcriptome analysis of OsJRL45 and OsJRL40 transgenic seedlings under salt stress, 3.3 Sequence differences between OsJRL45 and OsJRL40: marker-assisted selection (MAS) analysis for the selection of salt-tolerant rice. The revised manuscript this change can be found page number 17-20, line 325-423. Updated text in the manuscript if necessary. Thank you.

Reviewer 2 Report

Comments and Suggestions for Authors

This manuscript reported two novel genes OsJRL45 and OsJRL40 that can be used as molecular markers to select salt-tolerant rice species. The main findings are very interesting, and all experiments are well conducted. I have the following comments on this manuscript:

1.     The legend of Fig. 1 should be revised, as the authors also presented plant phenotypes and survival rate of rice species. In Fig. 1E, it seems that the expression of OsJRL45 in SR86 showed a decreasing trend after 6-48 h of salt treatment, is this trend significant? In addition, which tissue are the qPCR and RT-PCR results (Fig. 4 and Fig. 5E) obtained from? Roots, shoots or whole plants?

2.     The authors selected 9% NaCl solution (approximately 1.5 mol/L) to treat rice seedlings. Why did you use such a high salt treatment? In line 156-158, for F2 seedlings, you used 9% NaCl solution, but for F3 seedlings, why did you use 6% NaCl solution?

3.     A conclusion section is lacking in this version.

Author Response

Comments 1: The legend of Fig. 1 should be revised, as the authors also presented plant phenotypes and survival rate of rice species. In Fig. 1E, it seems that the expression of OsJRL45 in SR86 showed a decreasing trend after 6-48 h of salt treatment, is this trend significant? In addition, which tissue are the qPCR and RT-PCR results (Fig. 4 and Fig. 5E) obtained from? Roots, shoots or whole plants?

Response 1: The legend of Fig. 1 was revised to Phenotypic and analysis of OsJRL45 and OsJRL40 expression at the three-leaf stage under 9‰ NaCl salt stress. In Fig. 1E, the expression of OsJRL45 in SR86 showed a decreasing trend after 6-8h of salt treatment, but there was no significant downward trend from 8-48 hours. Tissue of the qPCR and RT-PCR results obtained from whole plants, such as Fig 1.E-F. The Fig. 4 and Fig. 5E (agarose gels) are from rice leaves. We have added this in the article Materials and Methods. The revised manuscript this change can be found page number 21, line 444 and line 463. Updated text in the manuscript if necessary. Thank you.

Comments 2: The authors selected 9% NaCl solution (approximately 1.5 mol/L) to treat rice seedlings. Why did you use such a high salt treatment? In line 156-158, for F2 seedlings, you used 9% NaCl solution, but for F3 seedlings, why did you use 6% NaCl solution?

Response 2: Thank you for pointing this out. We agree with this comment. The NaCl solution of rice seedlings under salt stress was 9‰, which has been corrected in the article, The revised manuscript this change can be found, thanks for pointing the error out. The F2 seedlings were subjected to 9‰ NaCl salt stress to select for salt-tolerant plants. Subsequently, the F3 seedlings were treated with a 6‰ NaCl solution, during which the phenotypic differences between salt-tolerant and salt-intolerant plants became particularly pronounced. Updated text in the manuscript if necessary. Thank you.

Comments 3: A conclusion section is lacking in this version.

Response 3: Thank you for pointing this out. We agree with this comment. The conclusion section of the paper has been added in the article. The revised manuscript this change can be found page number 22-23, line 498-518. Updated text in the manuscript if necessary. Thank you.

Round 2

Reviewer 1 Report

Comments and Suggestions for Authors

The authors responded to my comments and improved the manuscript sufficiently. The current version of the manuscript is acceptable for publication.